# Effects of Hydroxyapatite Additions on Alginate Gelation Kinetics During Cross-Linking

**DOI:** 10.3390/polym17020242

**Published:** 2025-01-19

**Authors:** Katarina Dimic-Misic, Monir Imani, Michael Gasik

**Affiliations:** 1Institute of General and Physical Chemistry, 11000 Belgrade, Serbia; 2Department of Chemical and Metallurgical Engineering, School of Chemical Engineering, Aalto University, 02150 Espoo, Finland; 3Department of Bioproducts and Biosystems, School of Chemical Engineering, Aalto University, 00076 Helsinki, Finland; monireh.imani@mirka.com; 4Mirka Ltd., Pensalavägen 210, 66850 Jeppo, Finland

**Keywords:** crosslinking kinetics, rheology, calcium alginate hydrogels, hydroxyapatite

## Abstract

Alginate hydrogels have gathered significant attention in biomedical engineering due to their remarkable biocompatibility, biodegradability, and ability to encapsulate cells and bioactive molecules, but much less has been reported on the kinetics of gelation. Scarce experimental data are available on cross-linked alginates (AL) with bioactive components. The present study addressed a novel method for defining the crosslinking mechanism using rheological measurements for aqueous mixtures of AL and calcium chloride (CaCl_2_) with the presence of hydroxyapatite (HAp) as filler particles. The time-dependent crosslinking behaviour of these mixtures was exploited using a plate–plate rheometer, when crosslinking occurs due to calcium ions (Ca^2+^) binding to the guluronic acid blocks within the AL polymer, forming a stable “egg-box” structure. To reveal the influence of HAp particles as filler on crosslinked sample morphology, after rheological measurement and crosslinking, crosslinked samples were freeze-dried and their morphology was assessed using an optical microscope and SEM. It was found that the addition of HAp particles, which are known to enhance the mechanical properties and biocompatibility of crosslinked AL gels, significantly decreased (usually rapidly) the interaction between the Ca^2+^ and AL chains. In this research, the physical “shielding” effect of HAp particles on the crosslinking of AL with Ca^2+^ ions has been observed for the first time, and its crosslinking behaviour was defined using rheological methods. After crosslinking and rheometer measurements, the samples were further evaluated for morphological properties and the observations were correlated with their dewatering properties. While the presence of HAp particles led to a slower crosslinking process and a more uniform development of the rheological parameters, it also led to a more uniform porosity and improved dewatering properties. The observed effects allow for a better understanding of the crosslinking process kinetics, which directly affects the physical and chemical properties of the AL gels. The shielding behaviour (retardation) of filler particles occurs when they physically or chemically block certain components in a mixture, delaying their interaction with other reactants. In hydrogel formulations, filler particles like hydroxyapatite (HAp) can act as barriers, adsorbing onto reactive components or creating physical separation, which slows the reaction rate and allows for controlled gelation or delayed crosslinking. This delayed reactivity is beneficial for precise control over the reaction timing, enabling the better manipulation of material properties such as crosslinking distribution, pore structure, and mechanical stability. In this research, the physical shielding effect of HAp particles was observed through changes in rheological properties during crosslinking and was dependent on the HAp concentration. The addition of HAp also enabled more uniform porosity and improved dewatering properties. The observed effects allow for a better understanding of the crosslinking process kinetics, which directly affects the physical and chemical properties of the AL gels.

## 1. Introduction

Alginate hydrogels (AL) are widely used in biomedical engineering due to their biocompatibility, biodegradability, and ability to encapsulate cells and bioactive molecules [1,2]. A successful deployment of AL gels requires careful control of key material properties, including biomechanical compliance, swelling behaviour, degradation rates, cell reactivity, and interactions with bioactive molecules and fillers [3,4,5]. By incorporating various fillers and employing diverse crosslinking techniques, it is possible to additionally fine-tune alginate-based biomaterials to meet specific needs [6,7,8].

Alginates are linear polyuronic polysaccharide extracted from brown seaweed (Phaeophycea) consisting of linked blocks of polymannuronic acid (M) and polyguluronic acid (G) with different sequential occurrence [9]. Recent macromolecular model investigations have demonstrated that divalent Ca^2+^ ions preferentially bind to the G -blocks, which are stiffer and have a more extended polymer chain compared to M blocks. The internal gelation technique occurs when Ca^2+^ ions trigger association of the polyguloronate sequences of the alginate chain by a dimerization mechanism, giving rise to aligned ribbon-like assemblies with cavities into which calcium ions are located, so called egg-box dimers [7,8]. As the extent of association increases through aggregation of the ordered dimers, clusters expand in size until they form a continuous three-dimensional crosslinking network [9].

AL gels crosslinked with divalent cations, of which calcium (Ca^2+^) is one of the most used, are used for different applications: biodegradable supports for water decontamination agents, scaffolds for cell culture, tissue engineering, drug delivery, and topical drug formulations [8,9]. Calcium-crosslinked AL hydrogels have been widely used in applications such as biodegradable supports for water decontamination agents, scaffolds for cell culture, tissue engineering, drug delivery, and topical drug formulations [10,11,12]. These materials offer key benefits, such as providing a moist environment that promotes wound healing and allowing for controlled drug distribution based on structure–activity relationships [13,14].

Crosslinked AL hydrogels have been utilized to fill voids in tissues in cases where bioactivity was not the primary goal. These hydrogels create a chemically favourable environment that supports essential physicochemical reactions [15,16,17]. For instance, composite beads have been developed using a combination of AL, hydroxyapatite (HAp), and a calcium chloride (CaCl_2_) solution [18,19,20]. Hybrid composite materials can enhance the strength and stiffness by incorporating inorganic fillers like hydroxyapatite. Hydroxyapatite shows great potential in tissue engineering applications due to its close resemblance to the inorganic component of human bone. This similarity endows it with unique properties, including biocompatibility, osteoconductivity, and the ability to promote bone cell adhesion and proliferation [21]. Its chemical similarity to natural bone enhances integration with host tissues, promoting efficient osteointegration (Figure 1) [22,23].

One of the methods to produce macroporous composite scaffolds based on AL and mineral particles comprises mixing of an alginate with particle dispersion in a CaCl_2_ solution followed by controlled slow gelation and subsequent freeze-drying [14,20]. Understanding the evolution and kinetics of crosslinking, directly impacting the mechanical properties and porosity, is critical for the fabrication of composite AL structures [15,16,17]. Porosity control is important, because fluid flow through scaffolds is important for successful fulfilment of complex parameters such as nutrient passage, cell growth, metabolic product removal and tissue regeneration [24].

It is common to use rheometers to study polymer and gel crosslinking kinetics: when crosslinking solutions are placed on the bottom plate of a rheometer and exposed to a crosslinking agent, they undergo a series of structural transformations [24,25]. These transformations can be tracked by monitoring the changes in their rheological parameters over time [25,26].

In addition, conventional rheometry relying on small amplitude oscillatory shear (SAOS) is limited by long measurement times, posing challenges when studying rapidly evolving materials [27,28]. While time sweeps at a constant frequency or strain can provide valuable data, they are often insufficient for capturing fast-evolving processes—especially for new material combinations without prior data about the proper operational window of the experimental parameters [29,30,31]. To address this limitation, researchers have implemented a combined frequency- and amplitude-strain method to catch crosslinking kinetics as a development of viscoelastic properties within a linear viscoelastic region (LVE) for these time-sensitive materials [27,30]. Using these methods, using viscoelastic measurements within a linear viscoelastic region, we have developed a novel measuring protocol that reveals the nonlinear crosslinking interval for various concentrations of HAp and Ca^2+^ ions and was able to access the porosity distribution using optical microscopy, SEM imaging, and the dewatering properties of samples. Using these methods, we developed a novel measurement protocol that utilizes viscoelastic measurements within the linear viscoelastic region. This approach allowed us to identify the nonlinear crosslinking interval for different concentrations of HAp and Ca^2+^ ions [26,27]. Additionally, we assessed the porosity distribution of the samples using optical microscopy, SEM imaging, and dewatering properties [32,33,34].

The present study proposes a novel method for defining the crosslinking mechanism using rheological measurements for aqueous mixtures of AL and calcium chloride (CaCl_2_) with the presence of hydroxyapatite (HAp) as filler particles. The present study provides novel insight into the following crosslinking mechanism using rheological measurements for aqueous mixtures of AL and calcium chloride (CaCl_2_) with the presence of hydroxyapatite (HAp) as filler particles. The time-dependent crosslinking behaviour of these mixtures was exploited using a plate-plate rheometer when crosslinking occurs due to calcium ions (Ca^2+^) binding to the guluronic acid blocks within the AL polymer, forming a stable “egg-box” structure. To reveal the influence of the concentration of Ca^2+^ ions, especially when HAp particles as filler were present, on the crosslinked sample morphology, crosslinked samples were freeze-dried and their morphology was assessed using optical microscope and SEM [35,36]. As more calcium ions become available, the pair-wise association monocomplexes form one-dimensional egg-box dimers, which aggregate via inter-cluster associations. In this way, rheology has been proposed for the observation of the dynamics of chemical reactions and cluster formation, and their consequences for the final crosslinked sample porosity. In this study, the objective was to track rheological parameters during the crosslinking interval, which in turn affects porosity and the related transport of fluids within these scaffolds [23,33]. The novel data in this work are related to the effect of delayed crosslinking when Ca^2+^ ions are in the presence of HAp particles, analysed with rheological measurements [34]. This allows for revealing the complex dynamics of crosslinking, which is linked to new possibilities in the control of porosity and the mechanical properties for such scaffold materials.

## 2. Materials and Methods

### 2.1. Preparation of Starting Solutions/Dispersions

Alginate solution was prepared by dissolving low-viscosity sodium alginate powder (A3249, AppliChem Darmstadt, Hamburg, Germany), which will be referred to in the text as alginate (AL) in distilled water, to obtain a 2% wt. concentration. Aqueous cross-linking solutions were prepared by dissolving CaCl_2_·2H_2_O powder (Sigma–Aldrich, Hamburg, Germany) in distilled water at concentrations of 0.3 and 0.4 wt.% calculated by dihydrate (which corresponds to 0.22 and 0.30% wt. of CaCl_2_ respectively). Hydroxyapatite (average particle size 2.5 µm, surface area ≥ 80 g·m^−2^; Sigma–Aldrich, Sant Luis, MO, USA) was added to these solutions at varying concentrations as shown below. SEM images of the HAp samples are presented in Figure 2.

Mixtures of crosslinking samples with varying concentrations of crosslinking solution CaCl_2_ and HAp are presented in Table 1. The percentages represent the amount of HAp filler in the CaCl_2_·2H_2_O dispersion and the percentage of AL solution in deionized water. These mixtures were prepared using a method based on similar experiments [10,32,33]. Experiments were conducted in several steps to define the effect of HAp on the gelation kinetics, morphology, porosity, liquid penetration and dewatering and mechanical properties of the scaffolds, as presented in Figure 3.

### 2.2. Rheological Measurements for Evaluation of SA Crosslinking

The kinetics of crosslinking and phase transformations of a 2% alginate (AL) solution was studied using a plate–plate rheometer (Anton Paar 302, Gratz, Austria). The SA solution (1.3 mL) was placed on the rheometer plate using a syringe. Crosslinking samples (1.3 mL), comprising CaCl_2_ solution and CaCl_2_–HAp dispersions (Table 1), were injected directly into the SA solution. These additions were made immediately before the start of each rheological measurement to observe the phase transformation process as presented in Figure 4.

Crosslinking dynamics were evaluated in the linear viscoelastic region (LVE). At the bottom of the rheometer, a Peltier temperature controller kept the temperature at 23 °C with the gap between plates at 1.5 mm. These conditions were used due to the presence of HAp particles causing possible strain hardening, known as “Mullins effect”—an increase in the viscoelastic properties and nonlinear argumentation within the crosslinked structure [35,36,37]. The linear viscoelastic region (LVE) was identified with oscillatory measurements. For amplitude sweep at *ω* = 0.1 rad·s^−1^, the critical strain was determined to be *γ*_c_ = 0.1%, and this was later used as a constant parameter at amplitude sweep measurements with the shear strain varying between *γ* = 0.01…500%. Following amplitude sweep tests, crosslinking dynamics were observed with rheological measurements in time-dependent mode with constant strain 0.1% and different constant angular frequencies (*ω* = 0.1, 0.5 and 1 rad·s^−1^) that simulated different mixing frequencies and probability of contacts between SA, Ca ions and HA particles. Once the viscoelastic parameters (transient complex viscosity (*η*^*+^), storage (*G*′) and loss modulus (*G*″)) reached their plateau values after 2500 s, the crosslinked films were removed from the rheometer. The measurements were performed five times.

### 2.3. Gravimetric Dewatering

Five crosslinked samples for each composite from Table 1 were removed from the rheometer and placed in an oven overnight at 60 °C, followed by placing the samples on a stuck of blotter papers, modifying the standard Åbo Akademi gravimetric dewatering measurement (ÅboGWR, Turku, Finland) for characterization of the water holding fine properties of the crosslinked AL gels [38]. A cylindrical vessel contained a filter made of polycarbonate membrane (Nucleopore Track-Etch membrane 5 μm, Whatman, Sigma–Aldrich, Sant Luis, MO, USA), below which were placed five crosslinked oven dried samples and five blotter papers as absorbents to avoid saturation. Water (5 mL) was added on top of the membrane and the overpressure of 0.5 bar was applied. After 105 s, the pressure was released and the weight of water that had passed through the blotter papers was measured and recorded. The blotter papers were weighed before and after the measurement, as presented in Figure 5. The weight difference was multiplied by 15,091 m^−2^, which is the inverse of the cylinder cross-sectional area. An average of five determinations was computed with the data variation found to be within ±10%.

### 2.4. Freeze-Drying of Samples

For analysis of the suitability of the samples crosslinked in the rheometer, the samples were removed from the rheometer bottom plate after the viscoelastic measurements of time-dependent behaviour, after 2500 s of constant strain γ = 0.01% and constant angular frequency ω = 0.1 rad·s^−1^ (assuming that with these parameters the structure of the sample was the least distorted) and dipped in liquid nitrogen for 3 min, as previously found to be an optimal time for freezing the total volume of the sample. After freezing, the samples were left for 24 h in a freeze-dryer (Labconco Freezone 2.5, Kansas City, MO, USA) at −50 °C and −24 kPa.

### 2.5. Optical and Scanning Electron Microscopy

Optical microscopy was applied to study the morphology of the crosslinked gel samples after the rheological measurements with an Olympus BX61 microscope equipped with a ColorView12 camera (Olympus, Shinjuku Monolith, 3–1 Nishi-Shinjuku 2-chome, Shinjuku-ku, Tokyo, Japan). Scanning electron microscopy (SEM) imaging was used to assess the morphology of the freeze-dried gel samples. The samples were coated with a thin layer of gold and then placed into a field emission scanning electron microscope (FE-SEM, model, Zeiss Sigma, Jena, Germany) with an accelerating voltage of 25 kV.

### 2.6. Reproducibility of Measurements

Data variation of the static water retention and rheometrical measurements for SA solutions were within ±5%, while for samples containing crosslinking solutions they were within ±10% due to the nature of the particle size of the sensitive gel structure. For the plate–plate geometry, there is also the presence of shear inhomogeneities contributing to reduced reproducibility. The data noise was partially removed with exponential smoothing [39].

## 3. Results and Discussion

Elastic moduli *G*′ and *G*″ are strain-dependent and with the increase in strain amplitude (γ) at a constant frequency (ω = 0.1 rad·s^−1^), the moduli change their magnitude due to the interactions of particles, ions and polymer chains. Crosslinking occurs when the storage modulus (*G*′) becomes equal to the loss modulus (*G*″), while the strain onset at which this occurs is called critical strain (γ_c_) [40]. This crosslinking strain moves towards higher values with the increase in Ca^2+^ concentration in the suspension, from 0.3 to 0.4 wt.% due to more interactivity and crosslinking segments with the alginate chains. An increase in HAp concentration widens the span and increases the values of γ_c_, the onset of the LVE region, defined as a value of *G*′ = 0.9 *G″*, allowing more time for the formation of a stable structure within the crosslinking gel [41]. The *G*′ value was found to decrease after LVE had been reached due to distortion of the viscoelastic structure, while the *G*″ value increased as a result of the effect of the gel-like structures observed previously and defined as “gel hardening” [42,43]. Crosslinking values of viscoelastic moduli *G*′ and *G*″ increased with an increase of the concentrations of both Ca^2+^ ions due to more sites for crosslinking and the physical presence of HAp particles that act as filler inserted into the alginate matrix (Figure 6c).

At a constant strain within the LVE region (*γ* = 0.1%) at a frequency sweep with *ω* = 0.01…100 rad·s^−1^, higher concentrations of Ca^2+^ led to increased values of both the shear storage modulus (*G*′) and the viscous modulus (*G*″). Samples containing 0.4% CaCl_2_ exhibited higher elastic moduli than those crosslinked with 0.3% CaCl_2_, as observed with amplitude sweep measurements. Also, increasing the amount of HAp particles further amplified the moduli (*G*′ and *G*″) due to a greater amount of sample volume filled with particles and their interactions that also caused agglomeration within the alginate matrix. For gels with lower concentrations of Ca^2+^ ions, *G*′ and *G*″, there were fewer crosslinking segments seen as the oscillatory dependence of the elastic moduli on deformation, i.e., *ω*, decreased, while the moduli were separated though the whole frequency range, without a distinct crosslinking of moduli (Figure 7a). Without HAp particles, moduli crossover at 1 rad·s^−1^ and a higher magnitude of *G*′ and *G*″ for samples with 0.4% CaCl_2_ was observed. As seen, when crosslinking was induced solely by a CaCl_2_ solution, without the presence of HAp particles, the increase in shear moduli (*G*′ and *G*″) was steeper and showed a more pronounced dependence on frequency [44,45].

Due to the oscillatory nature of the dependence of the viscoelastic parameters of the angular frequency, in order to compare the viscoelastic properties in respect to the presence of Ca^2+^ ions and the presence of Hap particles, in Table 2, we present the average values of the storage modulus (*G*′) and loss modulus (*G*″) (Figure 7) at distinct values of angular frequency (ω) 10, 50 and 100 rad·s^−1^, with slopes of fitted curves (*G*′ and *G*″ dependence on ω) with a power low model as an average of five measurements. With an increase of presence of Ca^2+^ ions, the slope of the curves increased from 0.3% to 0.4% due to a higher amount of crosslinking within the Al suspension matrix. With the increase of HAp concentration, the slope decreased due to a mechanism of physical shielding of Ca^2+^ ions by the HAp particles.

The effect of time-dependent structure development related to only interactions between AL, Ca ions and HAp, without external forces, was evaluated within LVE at ultralow and low constant strain (γ = 0.01% and γ = 0.1%, respectively), and at fixed angular frequencies (*ω* = 0.1, 1, 2.5, 5, and 10 rad·s^−1^). Increasing the rate of deformation enhanced particle-to-particle contact between HAp particles and a reduction of the physical barrier to enable expelling of Ca^2+^ ions towards the alginate molecular chains. Faster shearing through the suspension matrix, seen as strain of γ = 0.1%, in comparison to γ = 0.0.1%, accelerated both the crosslinking process and possibly the flocculation of the HAp particles, leading to higher shear moduli (*G*′ and *G*″) [46]. This rapid crosslinking created crosslinked zones that swelled, alongside areas that remained flat and non-crosslinked, causing flocculation in the gel matrix and an increase in the dynamic moduli [47]. For all samples, a lower concentration of HAp led to lower magnitudes of *G*′ and *G*″ and a pronounced nonlinear, oscillatory time-related pattern of increase. Slower motion of Ca^2+^ ions from the HAp particles, with lower values of ω, were caused by reduced shearing and collision between particles, resulting in a more linear, sinusoidal shape of the elastic moduli, particularly in samples containing 20% *w*/*w* HAp. An increase in Ca^2+^ concentration from 0.3% to 0.4% CaCl_2_ led to faster crosslinking and increased magnitudes of *G′* and *G*″ [48,49]. An increase of ω had a similar effect to raising the Ca^2+^ ion concentration, as it increased movement of HAp particles within the suspension matrix, related to a faster crosslinking rate. Increased collisions between HAp particles correlated with the movement of Ca^2+^ ions, resulting in swollen, crosslinked regions interspersed with flat, noncrosslinked areas of the alginate chains as described in previous research [50,51]. Additionally, raising the constant strain from 0.01% to 0.1% enhanced the thixotropic properties of the hydrogels, evidenced by the time-dependent behaviour of *G*′ and *G*’’. The average values of rheological data from Figure 8a–d is summarized in Figure 9a–c, presenting the average values from five measurements for two different strain levels (γ = 0.01% and γ = 0.1%, respectively). The elastic moduli (*G*′ and *G*″) increase is more prominent for higher strain (0.1%) due to the strain hardening effect and enhanced collision between HAp particles that enhance interactions between Ca^2+^ ions and AL polymer chains (Figure 8a–d), which increases crosslinking dynamics. Moreover, a progressive increase in sample deformation with an increase of the angular frequencies (ω) from 0.1 rad·s^−1^ to 5 rad·s^−1^ results in an increase in magnitude of G′ as a result of the faster aggregation and the stronger interactions between the AL and Ca^2+^ ions. Although the ion diffusion towards the alginate chains increases, more uniform crosslinking is achieved, contributing to the increased moduli [49,51].

The rheological data explicated in Figure 8a–d are summarized in Figure 9a–c, presenting average values from five measurements for two different strain levels (γ = 0.01% and γ = 0.1%, respectively). The elastic moduli (*G*′ and *G*″) increase is more prominent for higher strain (0.1%) due to the strain hardening effect and enhanced collisions between HAp particles that enhance interactions between Ca^2+^ ions and AL polymer chains (Figure 8a–d), which increases the crosslinking dynamics. Moreover, a progressive increase in sample deformation with an increase in the angular frequency (ω) from 0.1 rad·s^−1^ to 5 rad·s^−1^ resulted in an increase in magnitude of G′ as a result of the faster aggregation and the stronger interactions between the AL and Ca^2+^ ions. Although the ion diffusion towards the alginate chains increased, more uniform crosslinking was achieved, contributing to the increased moduli [49,51].

The time-dependent increase in transient, complex viscosity (*η*^*+^) at a constant strain of γ = 0.01% and a constant angular frequency ω = 0.1 rad·s^−1^ during crosslinking is presented in Figure 10 for suspensions containing HAp, which indicates the formation of a mechanically stronger structure, in comparison to AL suspensions crosslinked with only Ca ^2+^ ions [51]. The presence of HAp particles slows the rise in *η*^*+^, in contrast to the nearly immediate increase observed when Ca^2+^ ions are used as the sole crosslinking agent. The dynamic of *η*^*+^ increase is related to a reduced crosslinking rate, as fewer Ca^2+^ ions are available for crosslinking as they interact with HAp particles [32]. Furthermore, the addition of HAp particles reduces the rate of crosslinking by limiting the availability of free Ca^2+^ ions [36]. For solutions without HAp particles, the time required for suspensions to reach plateau values of *η*^*+^ is much shorter, with more time required when the concentration of Ca^2+^ ions is higher, most probably due to the thixotropic effect of strain hardening. As more calcium ions are available, aggregates of crosslinked guluronate sequencies increase via a nucleation growth mechanism, and as a consequence, the nuclei grow via coalescence of those gel nuclei, increasing transient complex viscosity. Conversely, the maximum value of transient complex viscosity (*η*^*+^_max_) is higher when HAp particles are present in the crosslinking suspension added to the SA solution, as shown in Figure 10c [52]. This suggests that while the crosslinking process may be slower with the presence of HAp particles, the resulting viscosity of the crosslinked gel matrix is higher, and the structure is more robust due to the incorporation of filler particles [53,54].

The average values of transient complex viscosity (*η*^*+^) within the LVE that follow the crosslinking with an increase in time, where they reach the plateau value and the onset of crosslinking (Figure 10) are presented in Table 3 as average values of five measurements.

The increase in static stress (*τ*_s_) during crosslinking, similarly to the development of transient complex viscosity, differs significantly depending on whether the crosslinking agent contains HAp particles or not. As presented in Figure 11a, rapid crosslinking with Ca^2+^ ions shows an immediate rise in static stress at the start of the measurement, with the “overshooting” character at the instant of the beginning of measurement [43]. This happens due to water being drawn into the “egg-box” crosslinked segments and then expelled, causing shrinkage and a drop in static stress, as shown in Figure 11a. Comparing the strength of the suspensions over time shows that the crosslinking kinetics depend on both the concentrations of Ca^2+^ ions and HAp particles. The static stress average values presented in Figure 11b show the average maximum static stress (τ^s^_max_) at plateau values at 2500 s, which has overshot for samples for which crosslinking occurred with Ca^2+^ ions, while the crosslinking is more gradual at higher HAp concentrations [54,55].

Numerical values of the average depicted in Table 4, calculated as average values of five measurements, show the differences of static stress values (*τ*_s_) at different times during crosslinking (Figure 11). The average value of measurements and yield static stress (τ_s_^0^) obtained as the plateau value of static stress are presented in Table 4, revealing the static stress values after 10, 500, 1000 and 2000 s, respectively, and from samples without HAp, due to the instantaneous junction of Ca. The offset of static stress, the sudden increase and decrease as the so-called “Mullins effect”, is seen as the overshoot is high for samples not containing HAp particles, and that the strength of the crosslinked hydrogels increases with the presence of HAp particles [49,54].

Optical images of crosslinked samples taken from the rheometer show that HAp particles tend to gather in the centre of the samples. In samples with lower HAp concentrations, crosslinking of AL happens more slowly in the centre, while the outer edges crosslink more quickly [50]. However, as the concentration of HAp increases, the particles distribute more evenly throughout the sample, leading to a more uniform crosslinking process. This results in smaller pores across the entire surface of the sample and a consistent colour. Figure 12 displays samples obtained during crosslinking at a constant static stress of γ = 0.1 s^−1^ and a constant angular frequency of ω = 0.1 rad·s^−1^. The samples show noticeable differences in colour: those without HAp appear transparent and porous with larger pores visible on the surface, while samples with HAp turn white as the pore size decreases, creating a denser appearance. Higher concentrations of HAp result in samples that are increasingly opaque and whiter.

Scanning electron microscopy (SEM) images were taken from freeze-dried crosslinked samples. Images are shown in Figure 13 for the samples with 0.3% CaCl_2_ and 0.4% CaCl_2_, with and without the presence of Hap particles in an Al solution matrix. The resulting SEM images reveal differences in aerogel porosity, showcasing a layered open structure with visible pores throughout the samples. When comparing samples crosslinked solely with CaCl_2_, where the diffusion of Ca^2+^ ions and crosslinking is faster, as observed rheologically, we observe larger pores with the pore size increasing for a higher amount of ions. The highly flocculated mechanism that results in large pores is suppressed with the presence of HAp, which exhibits slower crosslinking kinetics [47,48,49]. A higher concentration of Ca^2+^ ions promotes flocculation among the particles, leading to larger voids in the samples crosslinked only with Ca^2+^ ions. These voids arise from the rapid formation of junctions and the swelling of the egg-box structure, which traps water and creates pockets within the sample. Upon sublimation, these pockets leave behind larger empty voids in areas where more water was initially present.

In contrast, the incorporation of HAp enhances the development of a more compact structure with uniform porosity due to consistent internal stress during the crosslinking process. As illustrated in the optical images (Figure 13) and SEM images (Figure 13), the structure formation is influenced by both the kinetics of crosslinking and the presence of HAp. Additionally, smaller void sizes within the alginate layers that contain HAp are dependent on the concentration of HAp; as the concentration increases, the structure becomes denser and more compact.

Dewatering is directly linked to the porosity of the composite structure, evidenced by the flow of liquid through the porous material. An increase in Ca^2+^ ions accelerates the rate of crosslinking and enlarges the pores, while the presence of HAp reduces the pore size and increases the overall porosity, which usually improves the connectivity between the pores in the z-direction [55]. This connectivity is crucial for fluid transport and the transport of nutrients for tissue scaffolds and bone regeneration. Dewatering of filler-containing gel structures relies on strong particle–particle crowding interactions during the immobilization of Ca^2+^ ions and HAp particles throughout the crosslinking process [49,50]. After crosslinking and drying the samples in an oven, larger, swollen areas where junctions occur develop larger voids and pores, while areas where junctions have not occurred yet show smaller, more uniformly distributed pores [55,56]. When these oven-dried crosslinked samples are placed on blotter paper, they mimic the formation of a 3D scaffold intended for bone tissue regeneration. The arrangement of the crosslinked samples on top of each other was made to simulate the porosity in the z-direction within the scaffolds.

The dewatering behaviour of water samples between the stacked filter membranes, five crosslinked samples, and blotter papers varied based on the different gel strengths. The results presented in Figure 14 indicate that as the crosslinking kinetics decrease and the pore size becomes smaller with more pores, the dewatering rate of the stacked crosslinked samples decreases. This observation is supported by analysing the crosslinking time and gel strength, presented as *η*^*+^_max_ (Figure 10 and Table 3), the interval in which the transient complex viscosity reaches a plateau, and the corresponding dewatering values for varying concentrations of CaCl_2_ (0.3% and 0.4%) and Hap [53,57,58].

## 4. Conclusions

This study has explored the rheological behaviour of alginate gelation kinetics with calcium chloride and the resulting properties when hydroxyapatite is added to this system. This study seeks to highlight, in a novel way, the development of rheological parameters during the crosslinking of alginate with calcium ions, emphasizing the delayed reactivity caused by the shielding effect of hydroxyapatite particles (HAp). This delay in crosslinking, seen through rheological time dependent behaviour, which would otherwise occur immediately through egg-box structure formation, allowed for more uniform ion diffusion and loosely packed junctions between Ca^2+^ ions and alginate polymer chains. Consequently, the crosslinked samples exhibited reduced disparities between swelling and shrinking regions, resulting in more uniform porosity. The physical shielding effect of HAp particles on Ca^2+^ proved effective at two different calcium ion concentrations.

The presence of HAp particles led to a rheologically totally different crosslinking process than when there was free diffusion of Ca^2+^ ions towards AL polymer chains, which resulted in the introduction of new types of crosslinked hydrogels with varying porosity and fluid transfer properties. Complex viscoelastic behaviour was observed at various concentrations of CaCl_2_ and HAp in crosslinking dynamics, demonstrating the crosslinking process is significantly influenced by the concentration of HAp, which not only affects the kinetic rate of crosslinking but also alters the crosslinked material viscoelastic properties. This behaviour results in a more controlled and homogeneous structure within the crosslinked sample, with reduced pore sizes and enhanced connectivity among pores, which is essential for effective fluid transport. The SEM imaging and optical analysis confirmed that the presence of HAp contributes to a more compact structure with improved dewatering properties, facilitating better liquid flow through the samples.

The addition of HAp particles introduced a “shielding” effect that slowed the interaction between Ca^2+^ ions and the alginate chains, leading to prolonged crosslinking times. This behaviour resulted in a more controlled and homogeneous structure within the hydrogels, reducing the pore size and enhancing the connectivity among pores, which is essential for effective fluid transport. The SEM imaging and optical analysis confirmed that the presence of HAp contributes to a more compact structure with improved water retention properties.

The findings of this study emphasize the importance of adjusting both Ca^2+^ ion and HAp concentrations to optimize the viscoelasticity and porosity of alginate-based biomaterials. This research provides valuable insights for developing advanced scaffolds for tissue engineering applications, particularly in bone regeneration, where controlled porosity and mechanical strength are crucial. Overall, this study highlights the potential of using rheological techniques to better understand the crosslinking dynamics of alginate gels and their composite materials, enabling further development in the field of biomaterials.

## Figures and Tables

**Figure 1 polymers-17-00242-f001:**
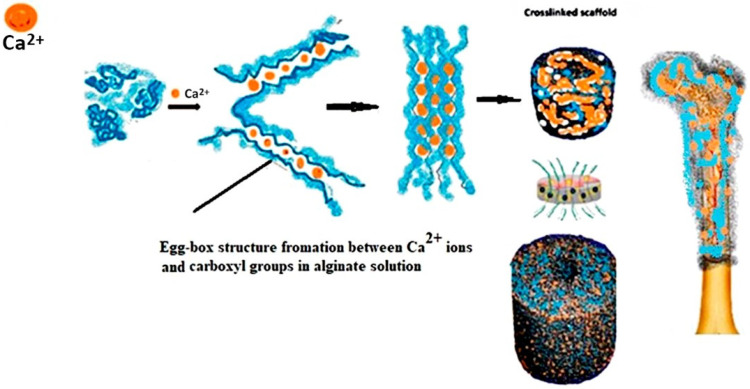
Schematic presentation of crosslinking of polyguloronate sequences of the alginate (AL) chain by a dimerization mechanism with Ca^2+^ ions via formation of the “egg-box” structure. Formed porous scaffolds made of crosslinked hydrogels enable bodily fluids transport through their matrix and can be used for tissue engineering and bone regeneration [5,14].

**Figure 2 polymers-17-00242-f002:**
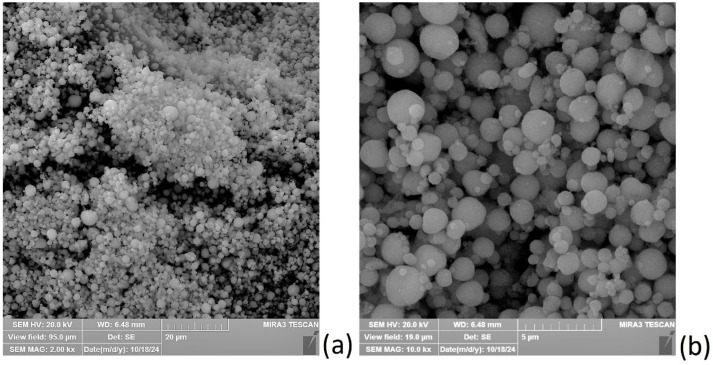
SEM images of hydroxyapatite (HAp) particles used in this research at two different magnifications, revealing particles of micrometre dimensions with a spherical structure (**a**,**b**).

**Figure 3 polymers-17-00242-f003:**
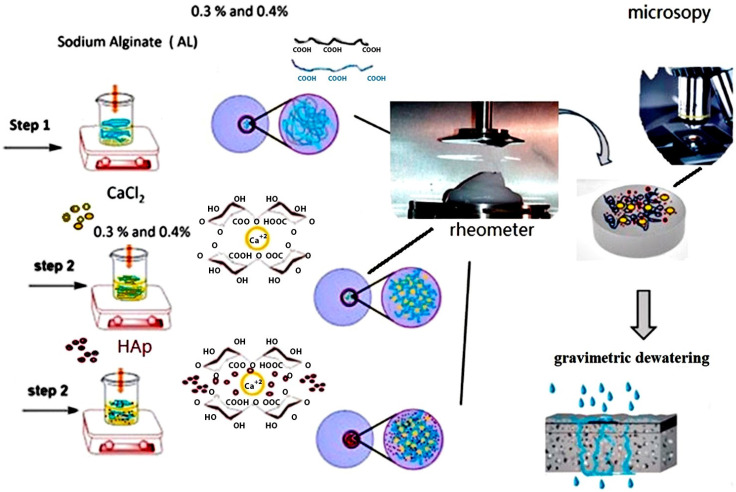
A schematic overview of the experimental setup in this study includes the preparation of suspensions, the molecular structure and formation of the egg-box network, rheological measurements leading to crosslinked structure formation, followed by SEM imaging and dewatering measurements.

**Figure 4 polymers-17-00242-f004:**
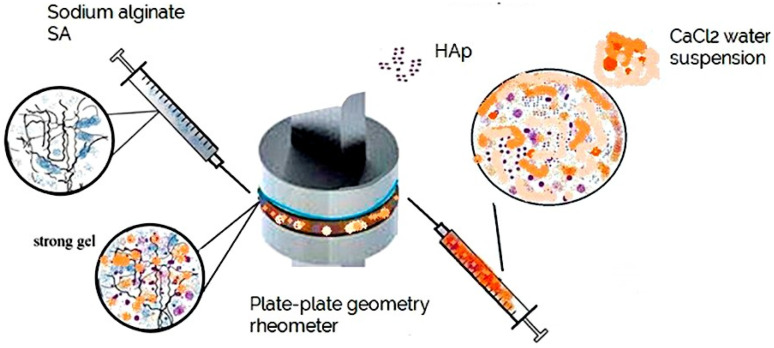
Schematic presentation of experimental set-up prior to rheological measurements; samples are mixed on the bottom plate of the rheometer prior to initiation of the measurements, leading to crosslinking of the AL solution with crosslinking dispersion of the suspension.

**Figure 5 polymers-17-00242-f005:**
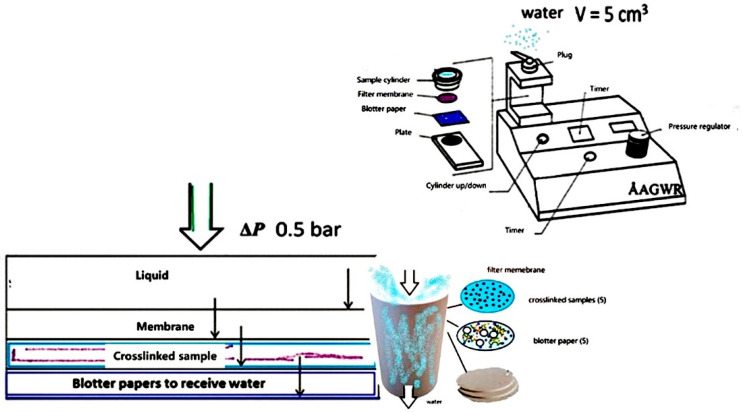
Schematic presentation of the experimental set up used for the gravimetric water retention device ÅA-GWR; crosslinked samples were placed between blotter papers and a porous membrane, influencing water retention.

**Figure 6 polymers-17-00242-f006:**
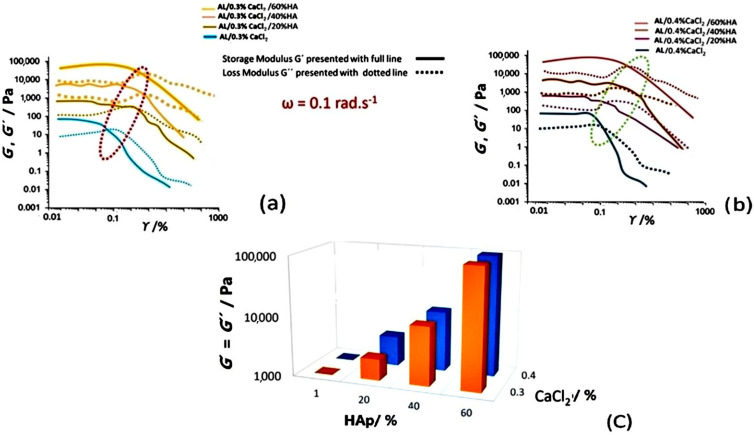
Viscoelastic properties of alginate (AL) and calcium chloride (CaCl_2_) crosslinking samples as a function of strain amplitude for constant angular frequency 0.1 rad·s^−1^ and increase in concentrations of HAp (**a**) 0.3% CaCl_2_, (**b**) 0.4% CaCl_2_ and (**c**) Calcium cloride CaCl_2_. Effect of concentration increase of CaCL_2_ and HAp on development of viscoelasticity and increase of elastic moduli as the crossover point where *G*′ = *G*″.

**Figure 7 polymers-17-00242-f007:**
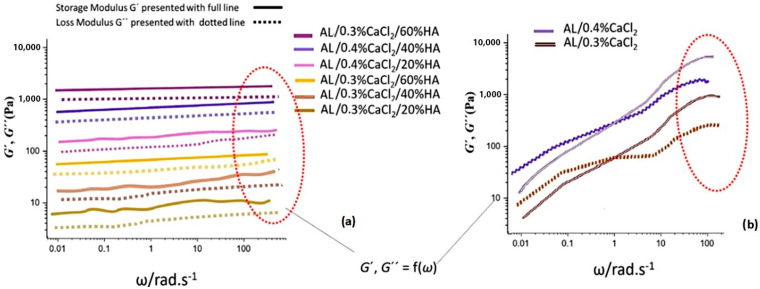
Frequency sweep results indicating reaction dynamic and crosslinking kinetics within the angular frequency (ω) range 0.01–150 rad·s^−1^ (**a**) for crosslinking suspensions containing both CaCl_2_ and HAp particles and (**b**) for crosslinking solutions containing only Ca^2+^ ions.

**Figure 8 polymers-17-00242-f008:**
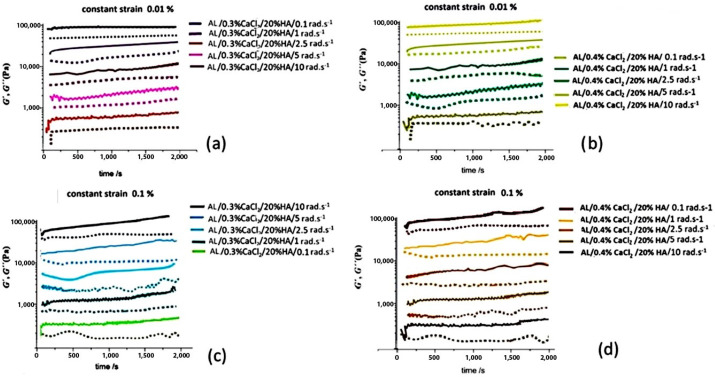
Elastic (*G*′) and loss (*G*″) moduli dependence on two constant strain (γ) values of 0.01% (**a**,**b**) and 0.1% (**c**,**d**) and constant angular frequencies (ω) of 0.1, 1, 2.5, 5 and 10 rad·s^−1^ for samples crosslinked with 0.3% CaCl2 (**a**,**c**) and 0.4% CaCl2 (**b**,**d**). Storage modulus (*G*′) full lines, loss modulus (*G*″) open lines.

**Figure 9 polymers-17-00242-f009:**
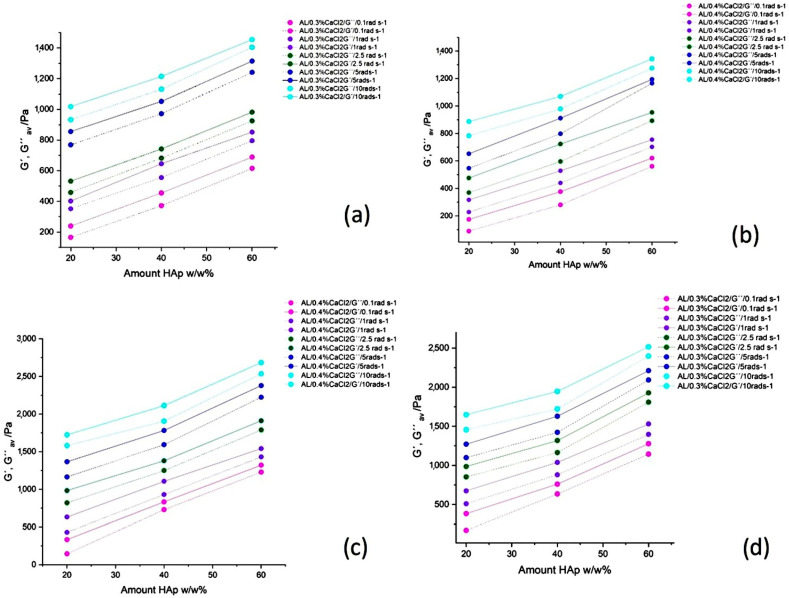
Viscoelastic moduli (*G*′ and *G*″)values for different constant strain values (γ = 0.01% (**a**,**b**) and γ = 0. 1% (**c**,**d**)) and different angular frequencies (**a**) *G*′ for concentration 0.3% CaCl_2_, (**b**) *G*″ for concentration 0.3% CaCl_2_, (**c**) *G*′ for concentration 0.4% CaCl_2_ and (**d**) *G*″ for concentration 0.4% CaCl_2_.

**Figure 10 polymers-17-00242-f010:**
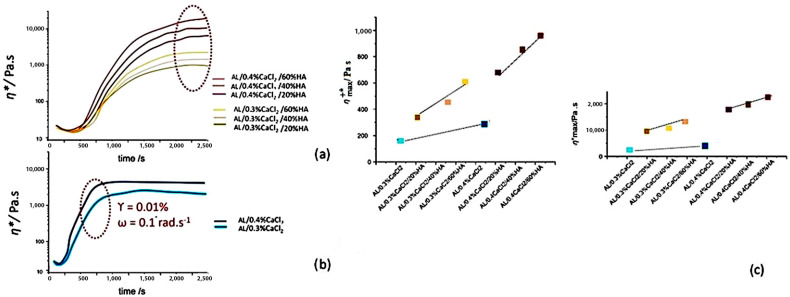
Crosslinked cluster formation during junction of calcium ions with alginate molecular chains presented with time-dependent increase of transient complex viscosity (*η*^*+^) increase up to the final crosslinked structure when the plateau value is reached (**a**) as a function of the concentration of CaCl_2_ and the presence of HAp particles, (**b**) gelation influenced only by the presence of Ca^2+^ ions and (**c**) maximum of transient complex viscosity *η*^*+^ _max_.

**Figure 11 polymers-17-00242-f011:**
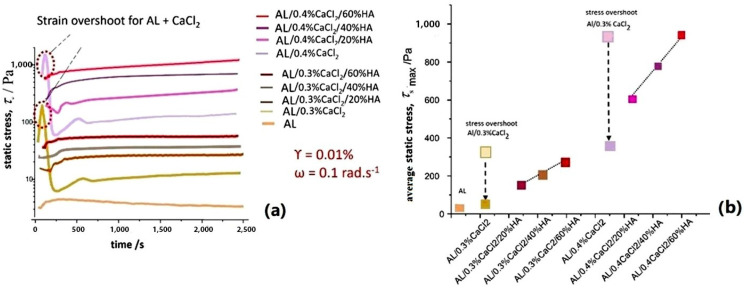
Structure formation during time dependent fluid gel crosslinking of alginate gels with calcium ions revealed with an increase in static stress (τ_s_). (**a**) Rheograms of static stress during crosslinking and (**b**) average values of maximum of reached static stress(*τ*_smax_), revealing the effect on overshoot due to instant monocomplex formation, containing compact swollen structures when suspended without HAp particles. An increase in calcium ion concentration and HAp particle concentration increases the crosslinking time.

**Figure 12 polymers-17-00242-f012:**
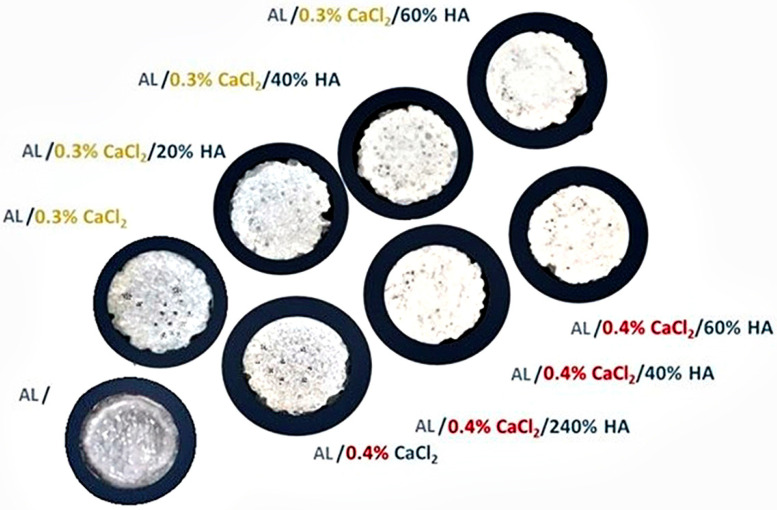
Optical camera images of the alginate samples after crosslinking as a function of calcium ion concentration and the amount of HAp filler particles. An increase of HAp filler particles increases the opacity and colour of the samples, becoming white with fewer large pores.

**Figure 13 polymers-17-00242-f013:**
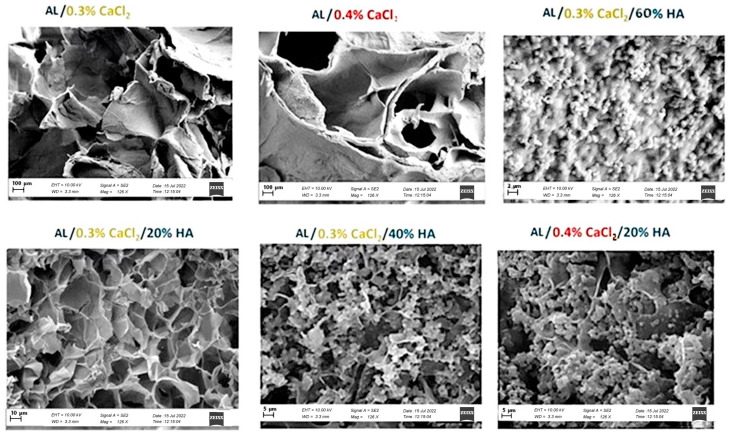
SEM images of samples after gelation. The difference in structure between more open layered pores with a lower amount of CaCl_2_ concentration, from 0.3 CaCl_2_ to 0.4% CaCl_2_, and more particles of Hap attached to alginate chains for 20% to 60% HAp, as revealed by a rheometer.

**Figure 14 polymers-17-00242-f014:**
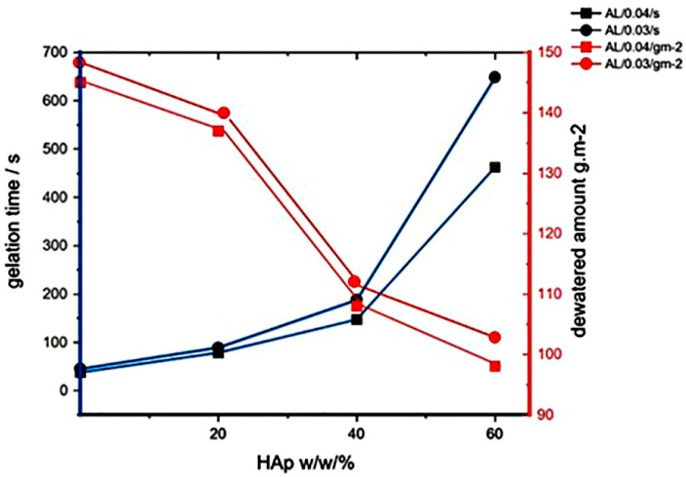
Correlation among gels with different concentration of CaCl_2_. Yield stress and dewatering properties of crosslinked samples as a function of the amount of HAp particles, using a ÅA GWR device for evaluation of the porosity of the crosslinked samples. The presence of HAp particles improves water retention and leads to a more uniform porosity, facilitating better liquid distribution through the samples.

**Table 1 polymers-17-00242-t001:** Crosslinking suspension composition and sample labelling used in this research.

CaCl_2_·2H_2_O wt.%	HAp wt.%	Sample Label
0.3	0	Al/0.3%CaCl_2_
	20	Al/0.3% CaCl_2_/20 HAp
	40	Al/0.3% CaCl_2_/40 HAp
	60	Al/0.3% CaCl_2_/60 HAp
0.4	0	Al/0.4% CaCl_2_
	20	Al/0.4% CaCl_2_/20 HAp
	40	Al/0.4% CaCl_2_/40 HAp
	60	Al/0.4% CaCl_2_/60 HAp

**Table 2 polymers-17-00242-t002:** Values of storage modulus (*G*′) and loss modulus (*G*″) at three distinct intervals of angular frequency (ω), representing behaviour at low (10 rad·s^−1^), intermediate (50 rad·s^−1^) and high values of angular frequency (100 rad·s^−1^).

Sample Label	*G*′_ω=10rad·s_^−1^/Pa	*G*″_ω=50rad·s_^−1^/Pa	*G*′_ω=50rad·s_^−1^/Pa	*G*″_ω=50rad·s_^−1^/Pa	*G*′_ω=100rad·s_^−1^/Pa	*G*″_ω=100rad·s_^−1^/Pa	*n_G_* _′_	*n_G_* _″_
AL/0.3CaCl_2_	58.6	62.6	63.2	68.3	101.5	92.7	5.3	6.5
Al/0.3CaCl_2_/20HAp	72.5	65.6	84.3	65.4	96.2	69.7	3.1	3.7
Al/0.3CaCl_2_/40HAp	355.3	314.8	397.1	325.8	425.4	379.1	2.8	3.3
Al/0.3CaCl_2_/60HAp	675.4	491.4	721.1	524.3	729.1	560.3	2.4	3.1
AL/0.4CaCl_2_	59.6	103.4	321.5	109.5	321.5	212.7	6.3	6.8
Al/0.4CaCl_2_/20HAp	1071.4	1097.3	1102.8	952.5	1102.3	1065.6	3.8	4.4
Al/0.4CaCl_2_/40HAp	3102.7	2913.4	3320.2	2913.4	3320.2	3187.7	3.4	3.7
Al/0.4CaCl_2_/60HAp	4821.6	4461.2	5168.3	4461.2	5168.3	4682.6	2.9	3.3

**Table 3 polymers-17-00242-t003:** Values of transient complex viscosity (*η*^*+^) during crosslinking at constant strain 0.1% and constant angular frequency 0.1 rad·s^−1^ for the given time intervals and maximum plateau value.

Sample Label	*η*^*+^_t=500s_/Pa·s	*η*^*+^_t=1500s_/Pa·s	*η*^*+^_t=2000s_/Pa·s	*η*^*+^_max_/Pa·s
AL/0.3CaCl_2_/	1748.5	1726.6	1754.3	1757.6
AL/0.3CaCl_2_/20HA	47.8	368.6	828.8	856.3
AL/0.3CaCl_2_/40HA	56.6	472.3	946.3	968.1
AL/0.3CaCl_2_/60HA	64.3	849.3	1178.7	1185.4
AL/0.4CaCl_2_/	5631.3	5724.6	5637.5	5729.5
AL/0.4CaCl_2_/20HA	49.1	1472.6	6489.3	6587.9
AL/0.4CaCl_2_/40HA	58.2	2574.3	7291.6	7358.3
AL/0.4CaCl_2_/60HA	69.3	6845.1	9745.4	9884.1

**Table 4 polymers-17-00242-t004:** Time dependent values of static stress (*τ*_s_) at distinct time intervals for all suspensions from Table 1.

Sample Label	*τ* _s_t= 10 s_/Pa_	*τ* _s_t= 500 s_/Pa_	*τ* _s_t= 1000 s_/Pa_	*τ* _s_t= 2000 s_/Pa_	*τ*_s_⁰/Pa
Al	7.5	5.4	4.2	4.7	4.7
AL/0.3CaCl_2_/	226.4	9.6	14.5	19.8	20.3
AL/0.3CaCl_2_/20HA	22.4	24.6	26.5	28.4	29.2
AL/0.3CaCl_2_/40HA	43.1	45.2	46.4	48.5	49.4
AL/0.3CaCl_2_/60HA	767.3	771.2	778.3	782.3	789.5
AL/0.4CaCl_2_/	1248.6	122.6	135.8	136.3	138.3
AL/0.4CaCl_2_/20HA	204.7	205.8	207.2	211.8	214.8
AL/0.4CaCl_2_/40HA	356.4	359.3	362.7	365.4	368.2
AL/0.4CaCl_2_/60HA	717.1	719.4	721.9	723.7	726.2

## Data Availability

The original contributions presented in this study are included in the article. Further inquiries can be directed to the corresponding author.

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
