# Peer review of "Effects of Hydroxyapatite Additions on Alginate Gelation Kinetics During Cross-Linking"

_polymers, 2025, doi:10.3390/polym17020242_

Round 1

Reviewer 1 Report

Comments and Suggestions for Authors

The authors investigated the crosslinking process for aqueous mixtures of sodium alginate and calcium chloride (CaCl₂) with hydroxyapatite (HAp), especially by rheology. Many studies are being conducted in the production of new materials, especially for the biomedical area. The manuscript needs a review before publication, especially in the following aspects:

>The authors commented "It was found that additions of HAp particles, which are known to enhance the mechanical properties and biocompatibility of crosslinked AL gels,...." However, the authors did not perform tests to evaluate biocompatibility and cytotoxicity;

>Page 2. Line 78-85. Authors should make clear the main contribution of the manuscript, in comparison with reports in the literature;

>Page 3. Preparation of starting solutions/dispersions. The criteria for producing the solution/dispersion are not clear, did the authors adopt previous literature? If so, report the reference;

>The authors indicated in the title "Effects of Hydroxyapatite Additions on Alginate Gelation Kinetics....". However, there was no test using any mathematical model, or the proposal of any new model.

Line 78. “In this study the objective was on analysis and evaluation of crosslinking kinetics….”.

>The authors could add FTIR results, in order to evaluate interactions or changes in chemical bands in the presence of Ca², HAp and alginate;

>Did the authors not perform any gel extraction tests to measure the degree of crosslinking?

>Page 13. “An increase in Ca² ions accelerates the rate of crosslinking and enlarges the pores, w…….”. Didn't the authors determine the average pore size of the analyzed samples? It would be better to add it in a graph or table;

>The authors should further explore the commented "shielding" behavior, especially its influence on possible application as a biomaterial;

Author Response

Answers to Reviewer 1

  1. The authors commented "It was found that additions of HAp particles, which are known to enhance the mechanical properties and biocompatibility of crosslinked AL gels,...." However, the authors did not perform tests to evaluate biocompatibility and cytotoxicity;

Answer:

In the Introduction section we mentioned that the main purpose of using Hap particles in composites for medical use is providing medical strength and stiffness where biocompatibility is not the main target and objective. HAp and alginate are used for decades as biocompatible materials which is a general knowledge. Hence there was no reason to study again biocompatibility of these materials. It is also noteworthy that biocompatibility such as under ISO10993 is mainly done for regulatory purposes and does not have a scientific added value (this is why e.g. Biomaterial journal rejects papers which use such biocompatibility testing).

We modified the sentences as follow: “Crosslinked AL hydrogels have been utilized to fill voids in tissues in cases where bioactivity was not the primary goal [15-17]. These hydrogels create a chemically favorable environment that supports essential physicochemical reactions [16]. For instance, composite beads have been developed using a combination of AL, hydroxyapatite (HAp), and a calcium chloride (CaCl₂) solution [19, 20]. Hybrid composite materials can enhance strength and stiffness by incorporating inorganic fillers like hydroxyapatite. Hydroxyapatite shows great potential in tissue engineering applications due to its close resemblance to the inorganic component of human bone. This similarity endows it with unique properties, including biocompatibility, osteoconductivity, and the ability to promote bone cell adhesion and proliferation [21].”

  1. Page 2. Line 78-85. Authors should make clear the main contribution of the manuscript, in comparison with reports in the literature;

Answer

This type of research has not been much used for sodium alginate crosslinking with Ca2+ ions but authors have used same methods for accessing viscoelastic properties of hydrogels such as with micro-nanocellulose. We added the following sentence:

 “Using these methods, we developed a novel measurement protocol that utilizes viscoelastic measurements within the linear viscoelastic region. This approach allowed us to identify the nonlinear crosslinking interval for different concentrations of HAp and Ca²⁺ ions [26, 27]. Additionally, we assessed the porosity distribution of the samples using optical microscopy, SEM imaging, and dewatering properties [30, 34]”.

  1. Page 3. Preparation of starting solutions/dispersions. The criteria for producing the solution/dispersion are not clear, did the authors adopt previous literature? If so, report the reference;

Answer

We made solutions of AL/water and HAp-water dispersion modifying methods described in ref. [16]. Evaluation of effect that Ca ions have on Al crosslinking and effect that presence of Hap has were presented combining methods from literature in references [10, 32, 33], but this is the first time to authors knowledge that precisely these concentrations were used. We modify sentences : “Solution of Al/water and HAp-water dispersion were made modifying method used in previous research [16]. Mixtures of crosslinking samples with varying concentrations of CaCl₂·2H₂O and HAp are presented in Table 1. The percentages represent the amount of HAp filler in the CaCl₂·2H₂O and AL solution in deionized water. These mixtures were prepared using method based on similar research [10, 32, 33].

  1. The authors indicated in the title "Effects of Hydroxyapatite Additions on Alginate Gelation Kinetics....". However, there was no test using any mathematical model, or the proposal of any new model.

Answer

We did not deploy any mathematical modeling but focused on experimental kinetics as the rate and progression of the gelation and crosslinking process, which are standard rheological procedures. These rheological methods are part of the rheometer testing protocols. We tried to understand the material's behavior during its transition from one state to another. Rheological measurements, as presented in Table 3, effectively capture changes in these properties, revealing the onset and progression of gelation. The term "gelation kinetics" in this research describes follow up of chemical reaction and its mechanism, as interpreted through rheological parameters. Hence the term "kinetics" here is defined by the parameters such as complex viscosity and viscoelasticity, changing over time during crosslinking.

  1. Line 78. “In this study the objective was on analysis and evaluation of crosslinking kinetics….”.

 Answer

We modify sentence in revised manuscript to : “In this study the objective was on tracking rheological parameters during crosslinking interval analysis and evaluation of development kinetics during crosslinking of AL with CaCl2-HAp dispersions, to assess the extent of introduction of a mineral component to the gel and to tune gelation process, which in turn affects porosity and related transport of fluids within these scaffolds [23,33].

  1. The authors could add FTIR results, in order to evaluate interactions or changes in chemical bands in the presence of Ca²⁺, HAp and alginate;

Answer

Authors thank reviewer from valuable comment, unfortunately in this study there was no option to use FTIR. Rather, as the scope of the paper is, we concentrated on physical, rheological observations and optical microscopy and morphological investigations of samples accompanied with dewatering properties.

  1. 7. Did the authors not perform any gel extraction tests to measure the degree of crosslinking?

Answer

Authors thank reviewer from valuable comment, but we did not perform gel extraction experiments.

  1. 8. Page 13. “An increase in Ca²⁺ ions accelerates the rate of crosslinking and enlarges the pores, w…….”. Didn't the authors determine the average pore size of the analyzed samples? It would be better to add it in a graph or table;

Answer

Pore size of samples were not calculated but presented qualitatively with optical imaging in Fig. 12, and SEM images at Fig. 13, and it has been demonstrated that they correlate with water retention properties of samples , as presented in Fig. 14., as in method mentioned in reference 34. This indicated clear dependence on pore size and porosity and water retention properties. During different rheological parameters, i.e. strain amplitude and angular frequency, pore size of samples would be different and therefore correlation was important to provide information about qualitative not quantitative results.

From practical point of view, water retention is an important feature as it connected not only with the total porosity but also tortuosity, permeability, permittivity and absorption which all are critical for a scaffold structure.

  1. The authors should further explore the commented "shielding" behavior, especially its influence on possible application as a biomaterial;

Answer:

We added the following sentences: “The shielding behavior (retardation) of filler particles occurs when they physically or chemically block certain components in a mixture, delaying their interaction with other reactants. In hydrogel formulations, filler particles like hydroxyapatite (HAp) can act as barriers, adsorbing onto reactive components or creating physical separation, which slows the reaction rate and allows for controlled gelation or delayed crosslinking. This delayed reactivity is beneficial for precise control over reaction timing, enabling better manipulation of material properties such as crosslinking distribution, pore structure, and mechanical stability. In this research, the physical shielding effect of HAp particles was observed through changes in rheological properties during crosslinking, dependent on HAp concentration. The addition of HAp also enabled more uniform porosity and improved dewatering properties.

Observed effects allows a better understanding of the crosslinking  process kinetics which directly affects physical and chemical properties of the AL gels.”

We hope these corrections will be sufficient for the consideration of the manuscript.

Reviewer 2 Report

Comments and Suggestions for Authors

I studied the manuscript “Effects of hydroxyapatite additions on alginate gelation kinetics during cross-linking” with great interest.

In my opinion, the manuscript does not meet the standards and quality required to be published in its current form. Actually, it is not very convincing and contains major flaws:

The manuscript should give more information about the theme as well as key results. The exact applications of the study findings are also not emphasized enough, which is insufficient to frame the whole picture of the present study.

Other comments:

-        The authors should distinguish between alginate and sodium alginate throughout the entire manuscript.

-        Figures were not well presented.

-        Results were not discussed enough.

Author Response

Answers to Reviewer 2

  1. The manuscript should give more information about the theme as well as key results. The exact applications of the study findings are also not emphasized enough, which is insufficient to frame the whole picture of the present study.

Answer

We have modified the conclusion section, and also we have added new sentences shown as follows:

This study seeks to highlight, in a novel way, the development of rheological parameters during the crosslinking of alginate with calcium ions, emphasizing the delayed reactivity caused by the shielding effect of hydroxyapatite particles (HAp). This delay in crosslinking, seen through rheological time dependent behaviour, which would otherwise occur immediately through egg-box structure formation, allowed for more uniform ion diffusion and loosely packed junctions between Ca²⁺ ions and alginate polymer chains. Consequently, the crosslinked samples exhibited reduced disparities between swelling and shrinking regions, resulting in more uniform porosity. The physical shielding effect of HAp particles on Ca²⁺ proved effective at two different calcium ion concentrations.

Presence of HAp particles lead to rheologically totally different crosslinking process than when there is free diffusion of Ca2+ ions towards AL polymer chains, which results in introduction of new types of crosslinked hydrogels with varying porosity and fluid transfer properties.”

  1. The authors should distinguish between alginate and sodium alginate throughout the entire manuscript.

Answer

We thank the reviewer for observing missing text and we make corrections. We modify sentence in Materials and Methods section: “Alginate solution was prepared by dissolving low-viscosity sodium alginate powder (A3249, AppliChem Darmstadt, Germany), that will be referred in text as alginate (AL) in distilled water to obtain a 2% wt. concentration.” In text we used term alginate and abbreviation AL.

We made changes throughout the manuscript and have used only term alginate which referred to sodium alginate.

  1. Figures were not well presented.

Answer: We improved the clarity and quality of figures. We particularly modified Fig. 3 to better present schematic of our experimental set up and mechanism of crosslinking.

  1. Results were not discussed enough.

Answer

”This crosslinking strain moves towards higher values with the increase in Ca 2+ concentration in the suspension, from 0.3 to 0.4 wt% due to more interactivity and crosslinking segments with alginate chains. Increase in HAp concentration widens the span and increases values of γc , onset of LVE region, defined as value of G´ = 0.9G”, allowing more time for formation of a stable structure within the crosslinking gel[41]. The G´ value was found to decrease after LVE has been reached due to distortion of viscoelastic structure, while the G´´ value increases as a result of effect of gel-like structures observed previously and defined as “gel hardening” [42]. Crosslinking values of viscoelastic moduli G´ and G´´ increase with increase of concentrations of both Ca 2+ ions due to more sights for crosslinking and physical presence of HAp particles that act as filler inserted into the alginate matrix (Fig. 6 c). Samples containing 0.4% CaCl₂ exhibit higher elastic moduli than those crosslinked with 0.3% CaCl₂, as observed with amplitude sweep measurements . Also increasing the amount of HAp particles further amplifies the moduli (G' and G'') due to greater amount of sample volume filled with particles and their interactions that also cause agglomeration within the alginate matrix. For gels with lower concentrations of Ca²⁺ ions, G´ and G´´, there is fewer crosslinking segments that is seen as oscillatory dependence of elastic moduli on deformation i.e. ω, whilst moduli are separated though the whole frequency range, without a distinct crosslinking of moduli (Fig. 7a).

“Due to oscillatory nature of dependence of viscoelastic parameters of angular frequency, in order to compare viscoelastic properties in respect to presence of Ca 2+ ions and presence of Hap particles, in Table 2 we present average values of Storage modulus (G´) and Loss modulus (G´´) (Fig. 7) at distinct values of angular frequency (ω) 10, 50 and 100 rad.s-1 , with slopes of fitted curves ( G´ and G´´ dependence on ω) with power low model as an average of five measurements. With increase of presence of Ca2+ions, from0.3% to 0.4% slope of curves increase due to higher amount of crosslinking within Al suspension matrix. With increase of Hap concentration slope decreases due to mechanism of physical shielding of Ca2+ ions with HAp particles. “

“The effect of time-dependent structure development related to only interactions between AL, Ca ions and HAp, without external forces, is evaluated within LVE at ultralow and low constant strain γ = 0.01% and γ = 0.1% respectively, and fixed angular frequencies (ω = 0.1, 1, 2.5, 5, and 10 rad·s⁻¹). Increasing the rate of deformation enhanced particle-to-particle contact between HAp particles and reduction of physical barrier as the enable expelling of Ca²⁺ ions towards the alginate molecular chains. Faster shearing through the suspension matrix , seen as strain of γ = 0.1%, in comparison to γ = 0.0.1% with accelerated both the crosslinking process and possibly the flocculation of HAp particles, leading to higher shear moduli (G´ and G´´) [46].”

“For all samples lower values led to lower magnitude of G´ nd G´´ and pronounced nonlinear, oscillatory time-related pattern of increase. Slower motion of Ca²⁺ ions from the HAp particles, for lower values of ω, caused by reduced shearing nd collision between particles resulted in more linear, slightly sinusoidal shape of the elastic moduli, particularly in samples containing 20% w/w HAp. An increase in Ca²⁺ concentration from 0.3% to 0.4% CaCl₂ led to faster crosslinking and increase in magnitude of G' and G'' [48,49]. Increase of ω had a similar effect to raising the Ca²⁺ ion concentration, as it increased movement of HAp particles within suspension matrix, related to faster crosslinking rate. Increased collisions between HAp particles correlate with the movement of Ca²⁺ ions, resulting in swollen, crosslinked regions interspersed with flat, uncrosslinked areas of the alginate chains as described in previous research [50,51]. Additionally, raising the constant strain from 0.01% to 0.1% enhanced the thixotropic properties of the hydrogels, evidenced by the time-dependent behaviour of G' and G.”

“The average values of rheological data from Figs. 8 a)-d) is summarized in Figs. 90 a)-c), present average values from five measurements for two different strain levels (γ = 0.01% and γ = 0.1%, respectively). The elastic moduli (G' and G'') increase is more prominent for higher strain (0.1%) due to strain hardening effect and enhanced collision between HAp particles that enhance interactions between Ca²⁺ ions and AL polymer chains (Figures 8)a-b and 8) c-d)), that increases crosslinking dynamics. Moreover, progressive increase in sample deformation with increase of the angular frequencies (ω) from 0.1 rad·s⁻¹ to 5 rad·s⁻¹ results in increase in magnitude of moduli as a result of the faster aggregation and the stronger interactions between the AL and Ca²⁺ ions. Although the ion diffusion toward the alginate chains increases, more uniform crosslinking is achieved, contributing to the increased moduli [49, 51].”

“Furthermore, the addition of HAp particles reduces the rate of crosslinking by limiting the availability of free Ca²⁺ ions [36]. For solutions without HAp particles, time required for suspensions to reach plateau values of η*+ is much shorter, with more time required when concentration of Ca 2+ ions is higher , most probably due to thixotropic effect of strain hardening. As more calcium ions are available aggregates of crosslinked guluronate sequencies increase via nucleation growth mechanism and as consequence nuclei grow via coalescence of those gel nuclei, increasing transient complex viscosity.

“Scanning electron microscopy (SEM) images were taken from freeze dried crosslinked samples. Images are shown in Fig. 13 for the samples with 0.3% CaCl2 and 0.4% CaCl2 , with and without presence of Hap particles in Al solution matrix. Resulting SEM images reveal differences in aerogel porosity, showcasing a layered open structure with visible pores throughout the samples. When comparing samples crosslinked solely with CaCl₂, where diffusion of Ca 2+ ions and crosslinking is faster, as observed rheologically, we observe larger pores with pore size increasing for higher amount of ions. Highly flocculated mechanism that results in large pores is suppressed with presence of HAp, which exhibit slower crosslinking kinetics[47-49].”

We hope these corrections will be sufficient for the consideration of the manuscript.

Reviewer 3 Report

Comments and Suggestions for Authors

It is an interesting manuscript about the effect of hydroxyapatite on the alginate gelation tracking dynamics in the cross linking. It merits to be published, however please consider the following suggestions: i) Please consider to mention the insight non studied previously in the abstract to highlight the novelty and maybe mentioning the difference of the finding in comparison to reported in literature. Ii) in the introduction in similar manner it could be highlighted the direction looking for new insights by the incorporation of a new source of calcium accompanied with a small organic molecule forming colloids that interact with the polymerization media. Please consider to highlight your research in comparison to previous ones justifying. Iii) In some schema it is recommended to add the chemical structures clearly to show the effect of molecular interaction and spatial distribution depending of functional groups, and different non-covalent interactions. iv) Check the need of further addition of information in the legends of figures in order to describe the different parts of the reaction flow. V) mention the reaction specific involucred in the cross linking. It is need it to imagine the process arriving in presence of the different possible media and how they affect the porosity. vi) about the materials and methods; question: in the experiments were taken different samples at varied times of the reaction to freeze? Mention some observation related if it is possible. vii) The results should show first the material and discuss about differences observed applying the methodology described. The discussion should focus on the development of molecular effect to form macroscopic material. So, it is recommended to discuss first these finding to then characterize the material with high impact physical-chemical parameters of the polymer. Viii) So, it is suggested to show first images to hen explain observation with the determination of rheological parameters.ix) In the case that authors don’t share the previous observations and it considered logic the order shown; unless add introductory redaction to results explaining the logic justifying the the determination of parameters to get information from the intrinsic matter to then show, describe and explain macroscopic observations.

Comments on the Quality of English Language

Please check comments to authors.

Author Response

Answers to Reviwer 3

  1. Please consider to mention the insight non studied previously in the abstract to highlight the novelty and maybe mentioning the difference of the finding in comparison to reported in literature.

Answer

In the abstract section we have modified sentences and written:

In this research physical "shielding" effect of HAp particles on crosslinking of AL with Ca2+ ions has been for the first time observed and crosslinking behavior defined using rheological methods. After crosslinking on rheometer samples were further evaluated morphologically has been observed and correlated with their dewatering properties. While presence of HAp particles led to slower crosslinking process and more uniform development of rheological parameters it influenced more uniform porosity and improved dewatering properties. Observed effects allows a better understanding of the crosslinking process kinetics which directly affects physical and chemical properties of the AL gels.”

In introduction we inserted sentences:

Alginates are linear polyuronic polysaccharide extracted from brown seaweed (Phaeophycea) consisting of linked blocks of polymannuronic acid (M) and polyguluronic acid (G) with different sequential occurrence [9]. Recent macromolecular model investigations have demonstrated that divalent Ca2+ ions preferentially bind to the G -blocks which are stiffer and have more extended polymer chain compared to M blocks. The internal gelation technique occurs when Ca2+ ions trigger association of the polyguloronate sequences of the alginate chain by dimerization mechanism, giving rise to aligned ribbon-like assemblies with cavities into which calcium ions are located, so called egg-box dimers [7-9]. At the extent of association increases through aggregation of ordered dimers, clusters expand in size untill they form a continuous three-dimensional crosslinking network [8,9].”

  1. In the introduction in similar manner it could be highlighted the direction looking for new insights by the incorporation of a new source of calcium accompanied with a small organic molecule forming colloids that interact with the polymerization media. Please consider to highlight your research in comparison to previous ones justifying.

Answer

In introduction, also as a part of answer to comments of another reviewer we add sentences:

“The present study addresses novel insight into following crosslinking mechanism using rheological measurements for aqueous mixtures of AL and calcium chloride (CaCl₂) with presence of hydroxyapatite (HAp), as filler particles. Time-dependent crosslinking behavior of these mixtures was exploited using a plate-plate rheometer, when crosslinking occurs due calcium ions (Ca²⁺) binding to the guluronic acid blocks within the AL polymer, forming a stable "egg-box" structure. To reveal the influence of concentration of Ca 2+ ions, especially when HAp particles as filler where present, on crosslinked sample morphology, crosslinked samples were freeze-dried and morphology was assessed using optical microscope and SEM [34, 36].As more calcium ions become available the pair wise association monocomplexes form one-dimensional egg-box dimerwhich aggregate via inter-cluster associations. In this way, rheology has been proposed for observation of dynamic of chemical reaction and cluster formations its consequence on final crosslinked sample porosity. In this study the objective was on tracking rheological parameters during crosslinking interval which in turn affects porosity and related transport of fluids within these scaffolds [23,33].”

  1. In some schema it is recommended to add the chemical structures clearly to show the effect of molecular interaction and spatial distribution depending of functional groups, and different non-covalent interactions.

Answer

We have modified some schematic presentation of measurements and mechanism of egg box structure in which divalent Ca2+ ions are trapped within alginate AL molecular chains (already explained in referred literature) and presence of HAp as presented in Fig. 3 (this also relects the answer to the comment No. 1). We believe that HAp particles present a physical barrier to Ca2+ ions in reaching Al molecular chains. We assume that readers already know basics of alginates as otherwise it will expand the paper beyond its major scope in rheology.

  1. Check the need of further addition of information in the legends of figures in order to describe the different parts of the reaction flow.

Answer

We improved quality of information in the figure legends and have written text as following:

Fig. 1. Schematic presentation of crosslinking of polyguloronate sequences of the alginate (AL) chain by dimerization mechanism with Ca2+ ions via formation of the “egg-box” structure. Formed porous scaffolds made of crosslinked hydrogels enable bodily fluids transport through their matrix and  can be used for tissue engineering and bone regeneration [5,14].

Fig.2 SEM images of hydroxyapatite (HAp) particles used in this research, at two different magnifications, revealing particles of micrometers dimensions with spherical structure: (a and b).

Fig.3 A schematic overview of the experimental setup in this study includes the preparation of suspensions, the molecular structure and formation of the egg-box network, rheological measurements leading to crosslinked structure formation, followed by SEM imaging and dewatering measurements.

Fig. 4.  Schematic presentation of experimental set-up prior to rheological measurements; samples mixing on bottom plate of the rheometer prior to initiation of measurements, that lead to crosslinking of AL solution with crosslinking dispersions suspensions.

Fig. 5 Schematic presentation of experimental set up used for Gravimetric Water Retention device ÅA-GWR; crosslinked samples were placed between blotter papers and porous membrane, influencing water retention.

Fig. 6. Viscoelastic properties of alginate ( AL) and calcium chloride ( CaCl2 ) crosslinking samples as a function of strain amplitude for constant angular frequency 0.1 rad·s-1. and increase in concentrations of HAp a) 0.3 % CaCl2  and b) 0.4% CaCl2. Effect of concentration increase  of CaCL2 and HAp on development of viscleasticity and increase of elastic moduli as crossover point where G´ = G´´.

Fig. 7  Frequency sweep results indicating reaction dynamic and crosslinking kinetic within the angular frequency (ω) range 0.01-150 rad.s-1 a) for crosslinking suspensions containing both CaCl2  and HAp particles and  b) for crosslinking solutions containing only Ca2+ ions.

Table 2 Values of storage modulus (G´) and loss modulus (G´´) at three distinct intervals od angular frequency  (ω), representing behavior at  low (10 rad.s-1), intermediate (50 rad.s-1)  and high values of angular frequency (100 rad.s-1).

Figure. 8 Elastic (G`) and loss (G´´) moduli dependence on two constant strain (γ) values of 0.01% (a, b) and 0.1% c, d)   and constant  angular frequencies (ω) of 0.1, 1, 2.5, 5 and 10 rad.s-1for samples crosslinked with 0.3%CaCl2 (a, c) and 0.4% CaCl2 (b, d). Storage modulus (G´) full lines, Loss Modulus (G´´) open lines.

Fig. 9 Viscoelastic moduli  (G´and G´´ )values for different constant strain values  strain  (γ= 0.01% a) and b) and γ= 0. 1%   c) and d)) and  different angular frequencies a) G´ for concentration 0.3 % CaCl2, b) G´´ for concentration 0.3 % CaCl2 c) G´ for concentration 0.4% CaCl2 and d)  G´´ for concentration 0.4% CaCl2

Fig.10   Crosslinked cluster formation during junction of calcium ions with alginate molecular chains presented with  time- dependent increase of  transient complex viscosity (η*+) increase up to final crosslinked structure when  plateau value is reached , a) as a function of  concentration of CaCl2 and presence of HAp particles b) gelation influenced only by the presence of Ca2+ ions and c) maximum of transient complex viscosity η*+ max.

Fig. 11.  Structure formation during time dependent fluid gel crosslinking of alginate gels with calcium ions revealed with increase in static stress (τs), a) rheograms  of static stress during crosslinking  and b) average values of maximum of reached static stress(τsmax), revealing effect on overshoot due to  instant monocomplexes formation containing compact swollen  structures when for suspension without HAp particles. Increase in calcium ions concentration and HAp particles concentration increases crosslinking time.  

Fig. 12. Optical camera images of the alginate samples after crosslinking as a function of  calcium ions concentration and amount of HAp filler particles. Increase of HAp filler particles increases opacity and colour of samples became white with fewer large pores.

Fig. 14  Correlation among gels with different concentration of CaCl2, yield stress and dewatering properties of crosslinked samples as a function of amount of HAp particles,  using ÅA GWR device for evaluation of porosity of crosslinked  samples. Presence of HAp particles improves water retention improves with more uniform porosity facilitating better liquid distribution through the samples.

  1. Mention the reaction specific involved in the cross linking. It is need it to imagine the process arriving in presence of the different possible media and how they affect the porosity.

Answer

We have presented crosslinking reaction between alginate polymer chains and Ca2+ ions in Fig. 1. We didn’t discuss in this manuscript other biopolymers neither other crosslinking mechanism, as the reactions might be quite complex and require more detailed studies. It is noteworthy that the scope of the paper remains in the rheological analysis of kinetics of cross-linking.

  1. About the materials and methods; question: in the experiments were taken different samples at varied times of the reaction to freeze? Mention some observation related if it is possible.

Answer

In Materials and Methods section we modified the paragraph related to freeze-drying in order to improve understanding of method:

“For analysis of suitability of samples crosslinked in the rheometer, samples were removed from the rheometer bottom plate after the viscoelastic measurements of time-dendent behaviour, after 2500 s of constant strain γ= 0.01% and constant angular frequency ω= 0.1 rad.s-1 ( assuming that with this parameters structure of sample was least distorted) and dipped in liquid nitrogen for 3 min, as previously found to be an optimal time for freezing the total volume of the sample. “

We established this method of rheological parameters to prepare freeze dried samples believing that they will demonstrate the structural influence of the crosslinking agent (Ca²⁺ ions) and the effect of HAp particles, without being impacted by external factors such as increased strain rates or angular frequency.

  1. The results should show first the material and discuss about differences observed applying the methodology described. The discussion should focus on the development of molecular effect to form macroscopic material. So, it is recommended to discuss first these finding to then characterize the material with high impact physical-chemical parameters of the polymer.

Answer

We have presented SEM images of HAp in Fig. 2. The molecular mechanism of egg-box structure formation during the crosslinking of alginate polymer (AL) with divalent ions such as Ca²⁺ has been extensively described in the literature. Therefore, we focused on observing the dynamics of the crosslinking mechanism. Since HAp is insoluble in water and AL at neutral pH, we propose that its role in the reaction is to act as a physical barrier, preventing Ca²⁺ ions from readily interacting with the AL polymer chains. In response to this observation and Reviewer 1's comments, we have revised the text accordingly.

In general, rheological studies do not provide a direct answer on a molecular effect of any similar reactions. Hence, we have only present a hypothesis which molecular interactions might be responsible for the behaviour of the materials under rheological loads.

  1. So, it is suggested to show first images to hen explain observation with the determination of rheological parameters.

9.In the case that authors don’t share the previous observations and it considered logic the order shown; unless add introductory redaction to results explaining the logic justifying the the determination of parameters to get information from the intrinsic matter to then show, describe and explain macroscopic observations.

 Answers to 8 and 9:

We structured our manuscript to follow a logical progression. First, we introduce how hydroxyapatite (HAp) is utilized for bone regeneration through its incorporation into the crosslinking mechanism of the egg-box structure. Next, we present a schematic overview of our methodology, emphasizing rheometry as the primary experimental technique, supported by morphological observations of samples crosslinked directly on the rheometer (with crosslinking monitored through the development of rheological parameters). Finally, we discuss dewatering measurements, aiming to uncover correlations between porosity, as observed via optical imaging and SEM microscopy, and the water retention properties of the samples.

We hope these corrections will be sufficient for the consideration of the manuscript.

Round 2

Reviewer 1 Report

Comments and Suggestions for Authors

The authors have satisfactorily addressed the questions, improving the quality of the manuscript and clarifying it. It has merit for publication in the revised version.

Reviewer 2 Report

Comments and Suggestions for Authors

The authors have implemented some alterations in the manuscript and improved it to a certain degree. However, it still does not meet the standards and quality required to be published in its current form. My previous revision remains.

Reviewer 3 Report

Comments and Suggestions for Authors

Thank you very much for your responses and modifications applied following suggestions. Now the manuscript is accepted for further editorial processing.